# Cryo-EM structure of the rhodopsin-Gαi-βγ complex reveals binding of the rhodopsin C-terminal tail to the gβ subunit

Ching-Ju Tsai[1†], Jacopo Marino[1†], Ricardo Adaixo[2†], Filip Pamula[1†], Jonas Muehle[1], Shoji Maeda[1‡], Tilman Flock[1,3], Nicholas MI Taylor[2§], Inayatulla Mohammed[2], Hugues Matile[4], Roger JP Dawson[4], Xavier Deupi[1,5]*, Henning Stahlberg[2]*, Gebhard Schertler[1,3]*

[1]Division of Biology and Chemistry / Laboratory of Biomolecular Research, Paul Scherrer Institute, Villigen, Switzerland; [2]Center for Cellular Imaging and NanAnalytics (C-CINA), Biozentrum, University of Basel, Basel, Switzerland; [3]Department of Biology, ETH Zurich, Zürich, Switzerland; [4]Pharma Research and Early Development, Therapeutic modalities, Roche Innovation Center Basel, Hoffmann-La Roche Ltd, Basel, Switzerland; [5]Condensed Matter Theory Group, Paul Scherrer Institute, Villigen, Switzerland

*For correspondence:
xavier.deupi@psi.ch (XD);
Henning.Stahlberg@unibas.ch (HS);
gebhard.schertler@psi.ch (GS)

†These authors contributed equally to this work

Present address: ‡Department of Molecular and Cellular Physiology, Stanford University School of Medicine, Stanford, United States; §Novo Nordisk Foundation Center For Protein Research, University of Copenhagen, Denmark, Europe

**Abstract** One of the largest membrane protein families in eukaryotes are G protein-coupled receptors (GPCRs). GPCRs modulate cell physiology by activating diverse intracellular transducers, prominently heterotrimeric G proteins. The recent surge in structural data has expanded our understanding of GPCR-mediated signal transduction. However, many aspects, including the existence of transient interactions, remain elusive. We present the cryo-EM structure of the light-sensitive GPCR rhodopsin in complex with heterotrimeric Gi. Our density map reveals the receptor C-terminal tail bound to the Gβ subunit of the G protein, providing a structural foundation for the role of the C-terminal tail in GPCR signaling, and of Gβ as scaffold for recruiting Gα subunits and G protein-receptor kinases. By comparing available complexes, we found a small set of common anchoring points that are G protein-subtype specific. Taken together, our structure and analysis provide new structural basis for the molecular events of the GPCR signaling pathway.
DOI: https://doi.org/10.7554/eLife.46041.001

## Introduction

G protein-coupled receptors (GPCRs) are the most diverse class of integral membrane proteins with almost 800 members in humans. GPCRs are activated by a great diversity of extracellular stimuli including photons, neurotransmitters, ions, proteins, and hormones (*Glukhova et al., 2018*). Upon activation, GPCRs couple to intracellular transducers, including four subtypes of G proteins (Gαs, Gαi/o, Gαq/11, Gα12/13) (*Milligan and Kostenis, 2006*), seven subtypes of GPCR kinases (GRKs) (*Gurevich et al., 2012*), and four subtypes of arrestins (*Smith and Rajagopal, 2016*) (*Figure 1A*), among many other partners (*Magalhaes et al., 2012*). While most GPCRs are promiscuous and can couple to more than one G protein subtype (*Flock et al., 2017*), the molecular determinants of G protein recognition are not yet fully understood. Understanding the molecular basis for G protein coupling and selectivity could lead to the design of drugs that promote specific signaling pathways and avoid unwanted side effects (*Hauser et al., 2017*).

The recent surge in the number of structures of GPCR-G protein complexes has greatly expanded our understanding of G protein recognition and GPCR-mediated signal transduction. Out of the 13 structures of GPCR-G protein complexes available, six contain a Gi/o subtype: μ-opioid receptor

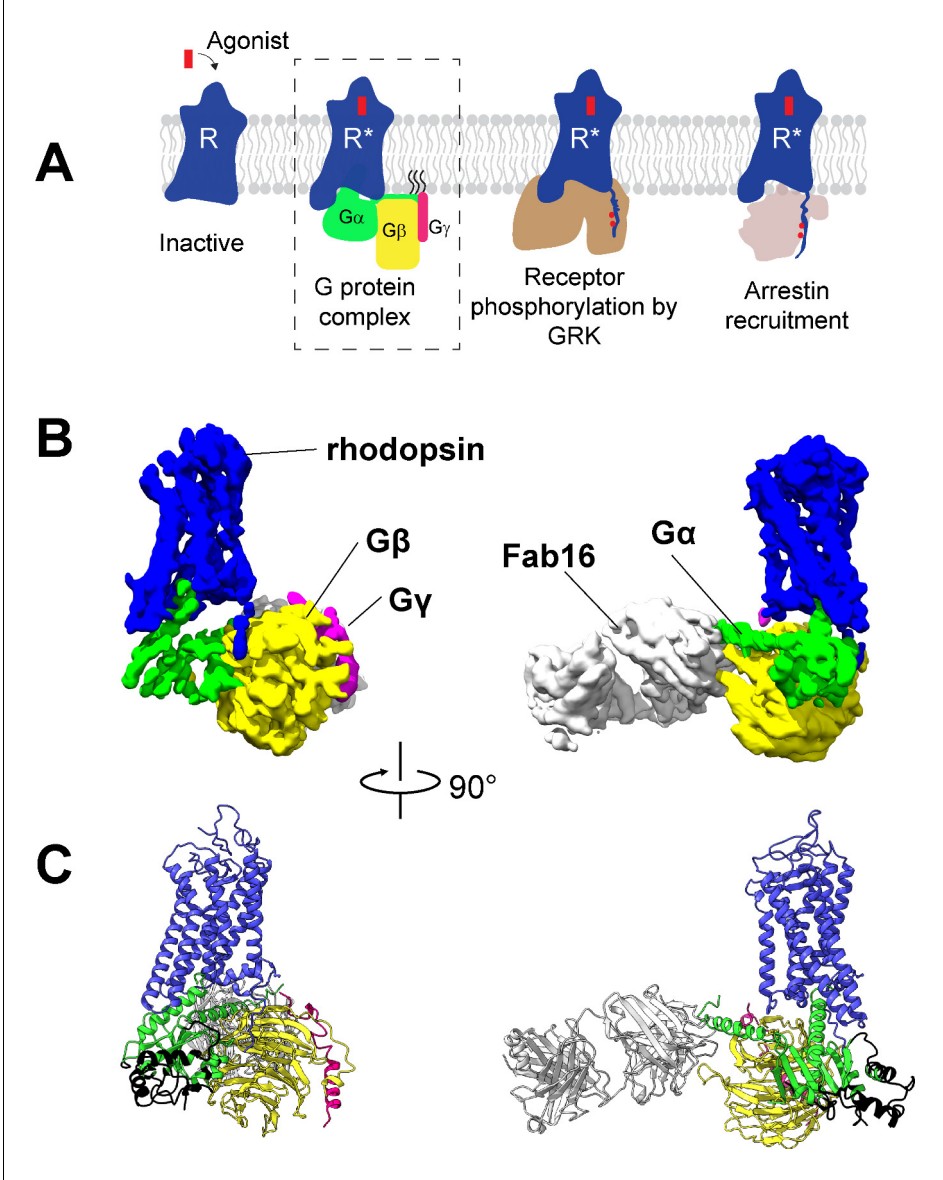

**Figure 1.** Cryo-EM structure of the rhodopsin-Gi-Fab16 complex. (**A**) GPCR signaling complexes. (**B**) EM density map of the complex (rhodopsin – blue, Gαi – green, Gβ – yellow, Gγ – magenta, Fab16 – white). (**C**) Atomic model of the complex (same color code as B). The region of the Ras domain of Gα with no corresponding density in the EM map is depicted in back.

DOI: https://doi.org/10.7554/eLife.46041.002

The following figure supplements are available for figure 1:

**Figure supplement 1.** Purification of the rhodopsin-Gi and rhodopsin-Gi-Fab16 complexes.
DOI: https://doi.org/10.7554/eLife.46041.003
**Figure supplement 2.** Cryo-EM maps of rhodopsin-Gi complexes with and without Fab16.
DOI: https://doi.org/10.7554/eLife.46041.004
**Figure supplement 3.** Image processing of the rhodopsin-Gi-Fab16 complex acquired with K2.
DOI: https://doi.org/10.7554/eLife.46041.005
**Figure supplement 4.** Image processing details.
DOI: https://doi.org/10.7554/eLife.46041.006
**Figure supplement 5.** 3D classification reveals the flexibility of the AH domain of Gαi.
DOI: https://doi.org/10.7554/eLife.46041.007
**Figure supplement 6.** Details of the cryo-EM density map of rhodopsin-Gi-Fab16 with a fitted atomic model.
DOI: https://doi.org/10.7554/eLife.46041.008

*Figure 1 continued on next page*

*Figure 1 continued*

**Figure supplement 7.** Comparison of the bovine rhodopsin-Gi complex to the other GPCR-G protein complexes.
DOI: https://doi.org/10.7554/eLife.46041.009

**Figure supplement 8.** Details of the source organism of the Gα, Gβ, and Gγ proteins used to obtain GPCR
G-protein complexes for structure determination.
DOI: https://doi.org/10.7554/eLife.46041.010

bound to Gi (*Koehl et al., 2018*), adenosine $A_1$ receptor bound to Gi (*Draper-Joyce et al., 2018*), cannabinoid receptor one bound to Gi (*Krishna Kumar et al., 2019*), human rhodopsin bound to Gi (*Kang et al., 2018*), $5HT_{1B}$ receptor bound to Go (*García-Nafría et al., 2018b*), and bovine rhodopsin bound to an engineered Go (mini-Go) (*Tsai et al., 2018*). However, the preparation of GPCR-G protein complexes for structural biology still remains challenging (*Munk et al., 2019*). Nanobodies (*Rasmussen et al., 2011*), Fab fragments (*Kang et al., 2018*; *Koehl et al., 2018*), and mini-G proteins (*García-Nafría et al., 2018b*; *Tsai et al., 2018*) have been very important tools to overcome the inherent instability and flexibility of these complexes and obtain near atomic-resolution structures. Importantly, these structures represent snapshots of a particular state of the complex in the signaling cascade, and therefore additional structural data are required to improve our understanding of this process (*Capper and Wacker, 2018*).

The photoreceptor rhodopsin is one of the best-characterized model systems for studying GPCRs, providing invaluable information on how receptor activation is translated into G protein and arrestin binding and activation (*Hofmann et al., 2009*; *Hofmann and Palczewski, 2015*; *Scheerer and Sommer, 2017*). Rhodopsin has been shown to interact with the Gβγ subunit of the G protein heterotrimer to assist binding and activation of the Gα subunit (*Herrmann et al., 2004*; *Herrmann et al., 2006*). After dissociation of the Gαβγ heterotrimer, the Gβ subunit recruits GRKs to the membrane, resulting in the phosphorylation of the receptor C-terminal tail (C-tail) (*Claing et al., 2002*; *Pao and Benovic, 2002*; *Pitcher et al., 1998*) and binding of arrestin (*Goodman et al., 1996*). Despite this biochemical evidence, a direct interaction between rhodopsin and Gβγ could not be observed in the existing complex (*Kang et al., 2018*).

Here, we present the cryo-EM structure of bovine rhodopsin in complex with a heterotrimeric Gi. Overall, our structure agrees well with current published structures (*Kang et al., 2018*; *Tsai et al., 2018*). Remarkably, the EM density map provides structural evidence for the interaction between the C-tail of the receptor and the Gβ subunit. The density map also shows that intracellular loops (ICL) 2 and 3 of rhodopsin are at contact distance to Gα. This prompted us to perform a comparison of all available structures of GPCR-G protein complexes to generate a comprehensive contact map of this region. We then extended this analysis to the binding interface formed by the C-terminal helix α5 of Gα and found that only a few G protein subtype-specific residues consistently bind to the receptors. These contacts are ubiquitous anchoring points that may be also involved in the selective engagement and activation of G proteins.

## Results

To obtain a rhodopsin-Gi complex suitable for structural studies, we expressed in HEK cells the constitutively active mutant of bovine rhodopsin N2C/M257Y/D282C (*Deupi et al., 2012*) which binds to the Gi protein heterotrimer (*Maeda et al., 2014*). The construct of the human Gαi1 subunit was expressed in *E. coli* (*Sun et al., 2015*), while the Gβ1γ1 subunit was isolated from bovine retinae and thus contains native post-translational modification important for transducin function (*Matsuda and Fukada, 2000*). We reconstituted the rhodopsin-Gαi1β1γ1 complex in the detergent lauryl maltose neopentyl glycol (LMNG) (*Maeda et al., 2014*) in presence of Fab16 (*Maeda et al., 2018*) (*Figure 1—figure supplement 1*), as cryo-EM screening revealed that the complex without Fab could not be refined to high resolution (*Figure 1—figure supplements 2* and *3*). During image processing (*Figure 1—figure supplements 3* and *4*), 3D classification revealed that the density corresponding to Gα is heterogeneous (*Figure 1—figure supplement 5*), particularly at flexible regions such as the α-helical (AH) domain. The AH domain was then excluded by using a soft mask during refinement, resulting in a map with a nominal global resolution of 4.38 Å (*Figure 1—figure supplement 3D and*

*E*). The EM map was used to build a model of the complex (*Figure 1*, *Figure 1—figure supplement 6*).

## Architecture of the rhodopsin-G protein complex

The structure of rhodopsin-Gi-Fab16 reveals the features observed in previously reported GPCR-G protein complexes (*Figure 1B and C*; *Figure 1—figure supplement 7A*). In particular, our cryo-EM structure is in excellent agreement with the crystal structure of the same rhodopsin mutant bound to a mini-Go protein (*Tsai et al., 2018*), with a nearly identical orientation of the C-terminal α5 helix (*Figure 1—figure supplement 7B*), which contacts transmembrane helices (TM) 2, 3, 5–7 and the TM7/helix 8 (H8) turn of the receptor (*Figure 1—figure supplement 6B*).

## Interaction between the C-terminal tail of rhodopsin and Gβ

The EM map reveals a density on the Gβ subunit as continuation of H8 of the receptor (*Figure 2A*), which corresponds to the C-tail of rhodopsin. This feature has only been observed in the recent structure of the M1 muscarinic acetylcholine receptor in complex with G11 (*Maeda et al., 2019*). We modeled half of the C-tail of rhodopsin (12 out of 25 residues; 324–335) into this density as a continuous stretched peptide with residues G324, P327 and G329 serving as flexible hinges (*Figure 2B*). This allows us to compare the structure of this region in three conformational states of the receptor: inactive (*Okada et al., 2004*), G protein-bound (this structure), and arrestin-bound (*Zhou et al., 2017*) (*Figure 2B*).

In the inactive state, the C-tail folds around the cytoplasmic side of rhodopsin, although this is likely due to crystal packing as this region is intrinsically disordered in the absence of a binding partner (*Jaakola et al., 2005*; *Venkatakrishnan et al., 2014*).

In our G protein complex, the C-tail stretches over a polar surface on the cleft between Gα and Gβ, interacting with both subunits (*Figure 2C and D*). In Gβ, the interacting residues are C271, D290, and D291 in blade six and R314 in blade 7 of the β-propeller (*Figure 2D*), which impart a local negative electrostatic potential (*Figure 2—figure supplement 1*). These residues are conserved in Gβ1–4 (*Figure 2—figure supplement 2B*) but not in Gβ5, the least similar to the other Gβ subtypes (*Dupré et al., 2009*). In Gα, the interacting residues with the C-tail are N256H3.15 and K257H3.16 in helix 3 of the Ras domain (*Figure 2D*), which confer instead a local positive electrostatic potential (*Figure 2—figure supplement 1*). These residues are quite conserved across G protein subtypes except Gq (*Figure 2—figure supplement 2A*). Interestingly, these regions in Gα and Gβ that contact the receptor C-tail are also involved in recognition of GRK2 (*Tesmer et al., 2005*).

In the rhodopsin-arrestin complex part of the proximal segment of the C-terminus (residues 325–329) is not resolved, but the distal part could be modeled up to S343, eight residues longer than in our G protein complex and including two phosphorylated sites. In the presence of arrestin, the C-tail stretches further, with residues K339-T342 forming a short β-strand antiparallel to the N-terminal β-strand of arrestin (*Figure 2B*).

Thus, it appears that distinct portions of the C-tail are responsible for contacting different intracellular partners. The central part of the C-tail (residues 330–335) can bind to Gα and Gβ, while the distal half of the C-tail (residues 336–343), which contains five (out of six) phosphorylation sites (*Azevedo et al., 2015*), binds to arrestin (*Figure 2C*).

## Comparison of the GPCR-G protein binding interface

As in the other existing GPCR-G protein complex structures, the C-terminal α5 helix of Gα forms the major contact interface to rhodopsin. The α5 helix consists of the last 26 amino acids of Gα (H5.01–26), in which the last five residues fold into a hook-like structure (*Tsai et al., 2018*). The majority of the contacts formed by the α5 helix to GPCRs concentrate in the region from H5.11 to H5.26 (*Glukhova et al., 2018*) (*Figure 3—figure supplement 1*).

We aligned the structures of available complexes using the Cα atoms of H5.11–26. This alignment reduces apparent differences in the binding positions and provides the 'viewpoint of the G protein' (*Figure 3A*). We compiled an exhaustive list of the residue-residue contacts between the receptor and the α5 helix in all the available GPCR/G protein complexes, and observed that the main contacts to the α5 helix are formed by TM5 and TM6, followed by TM3 and TM7/8 (*Figure 3B*, *Figure 3—figure supplement 2*).

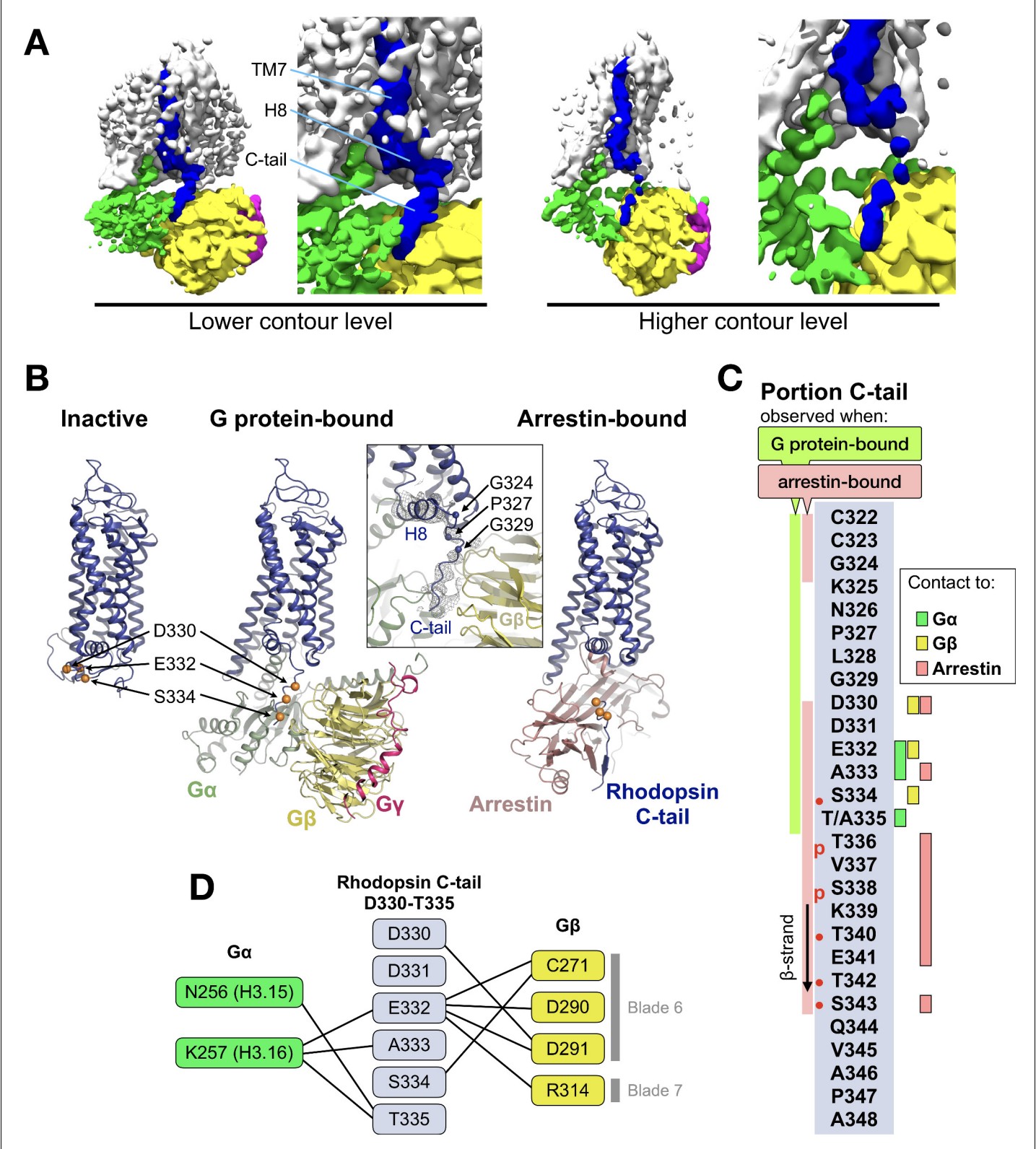

**Figure 2.** The C-terminal tail of rhodopsin. (A) The EM map is contoured at two different levels to show the continuity of the density. The weakening at the end of H8 may arise from impaired interactions of the receptor with the detergent micelle (*Glukhova et al., 2018*). TM7, H8 and the C-tail of the receptor are colored in blue, Gα in green, Gβ in yellow, and Gγ in magenta. (B) Conformational change of the C-tail between three different conformational states of rhodopsin: Inactive state (left, PDB id: 1U19), G protein-bound (center, this work), and arrestin-bound (right, PDB id: 5W0P,

*Figure 2 continued on next page*

*Figure 2 continued*

chain A). The Cα atoms of residues Asp330, Glu332 and Ser334 are shown as orange spheres to help tracking the structural changes in the C-tail. All structures are aligned to rhodopsin. (C) Schematic representation of the rhodopsin C-tail from Cys322 to Ala348. On the left, colored bars indicate the portion of the C-tail visible in this structure (green), and in the arrestin-bound structure (salmon) (PDB id: 5W0P). On the right, the residue-residue contacts between rhodopsin C-tail and Gα (marked in green), Gβ (yellow), and arrestin (salmon) within 4 Å distance are indicated. Thr336 and Ser338 are phosphorylated in the arrestin-bound structure. The predicted phosphorylation sites are marked with red dots. (D) Model of residue-residue interaction between the rhodopsin C-tail and the G protein subunits. Asn$^{H3.15}$ and Lys$^{H3.16}$ of the Gαi subunit forms the contact to the C-tail of rhodopsin near Glu332, Ala333 and Thr 335. In this model, the surface region of blades 6 and 7 of Gβ contact the C-tail via hydrophilic residues Cys271, Asp290, Asp 291, and Arg314.

DOI: https://doi.org/10.7554/eLife.46041.011

The following figure supplements are available for figure 2:

**Figure supplement 1.** Electrostatic potential.

DOI: https://doi.org/10.7554/eLife.46041.012

**Figure supplement 2.** Sequence conservation in Gα and Gβγ.

DOI: https://doi.org/10.7554/eLife.46041.013

**Figure supplement 3.** Flexible fitting of the C-tail in the electron density.

DOI: https://doi.org/10.7554/eLife.46041.014

**Figure supplement 4.** Structure of the C-tail in rhodopsin and M1R.

DOI: https://doi.org/10.7554/eLife.46041.015

Gi/o-bound complexes have two conserved contacts: one between Gly$^{H5.24}$ and the TM7/H8 turn, and one between Tyr/Phe$^{H5.26}$ and TM6. In Gs-bound complexes, three distinct interactions are found: between Arg$^{H5.12}$ and TM3/ICL2, and between Leu$^{H5.26}$ and Arg$^{H5.17}$ and TM5 (*Figure 3C,D*). In Gi/o complexes, the equivalent Lys$^{H5.17}$ lies instead roughly parallel to TM6. Interestingly, Arg$^{H5.17}$ appears to consistently interact with residue 5.68 in class A GPCRs, and with Lys$^{5.64}$ in class B GPCRs (*Figure 3E*, *Figure 3—figure supplement 2*). We suggest that a tight interaction between Arg$^{H5.17}$ and TM5 might be one of the main determinants for the Gs-specific relocation of TM6.

Besides the canonical interaction with the α5 helix, our complex shows that ICL2 and ICL3 of the receptor are at contact distance to Gi (*Figure 3—figure supplement 3A*). In all analyzed structures, we found that ICL2 lies near αN/β1 and β2/β3 of Gα, while ICL3 is close to α4/β6 (*Figure 3—figure supplements 3* and *4*). Interestingly, ICL2 folds into an α-helical structure in all the class A receptors except rhodopsin (*Figure 3—figure supplement 5*).

ICL2 does not contribute to binding Gα in the structures of 5HT$_{1B}$-mini-Go and A$_{1A}$R-Gi. Nevertheless, the contact between Gα and ICL2 seems to discriminate between class A GPCRs –which interact via the αN/β1 loop– and class B GPCRs –which instead the region around β1 and β2/β3 (*Figure 3—figure supplement 4*).

ICL3, one of the most diverse regions in GPCRs, is often not completely resolved in the available structures (*Figure 3—figure supplement 3D*), either because it is too flexible or because it has been engineered to facilitate structural determination (*Munk et al., 2019*). Gi/o-coupled receptors use residues on TM5/ICL3/TM6 to contact the α4/β6 region, while Gs-coupled receptors mainly use TM5/ICL3 (*Figure 3—figure supplement 4*). This difference is due to the larger displacement of TM6 in Gs complexes. In ICL3, residue Tyr$^{G.S6.02}$, highly conserved in all G protein subtypes, engages the receptor in about half of all the complexes. This residue has been shown to be crucial for the stabilization of the rhodopsin-Gi complex (*Sun et al., 2015*).

Thus, our analysis suggests that ICL2 and ICL3 contribute to the binding interface between receptor and G protein, a feature that may be required to further stabilize the complex during nucleotide exchange.

## Discussion

In this work, we present the cryo-EM structure of the signaling complex between bovine rhodopsin and a Gi protein heterotrimer, stabilized using an antibody Fab fragment (Fab16) (*Maeda et al., 2018*). Overall, this structure agrees very well with existing complexes. In particular, we found that the binding mode of the G protein Ras domain –including the key C-terminal α5 helix– is virtually

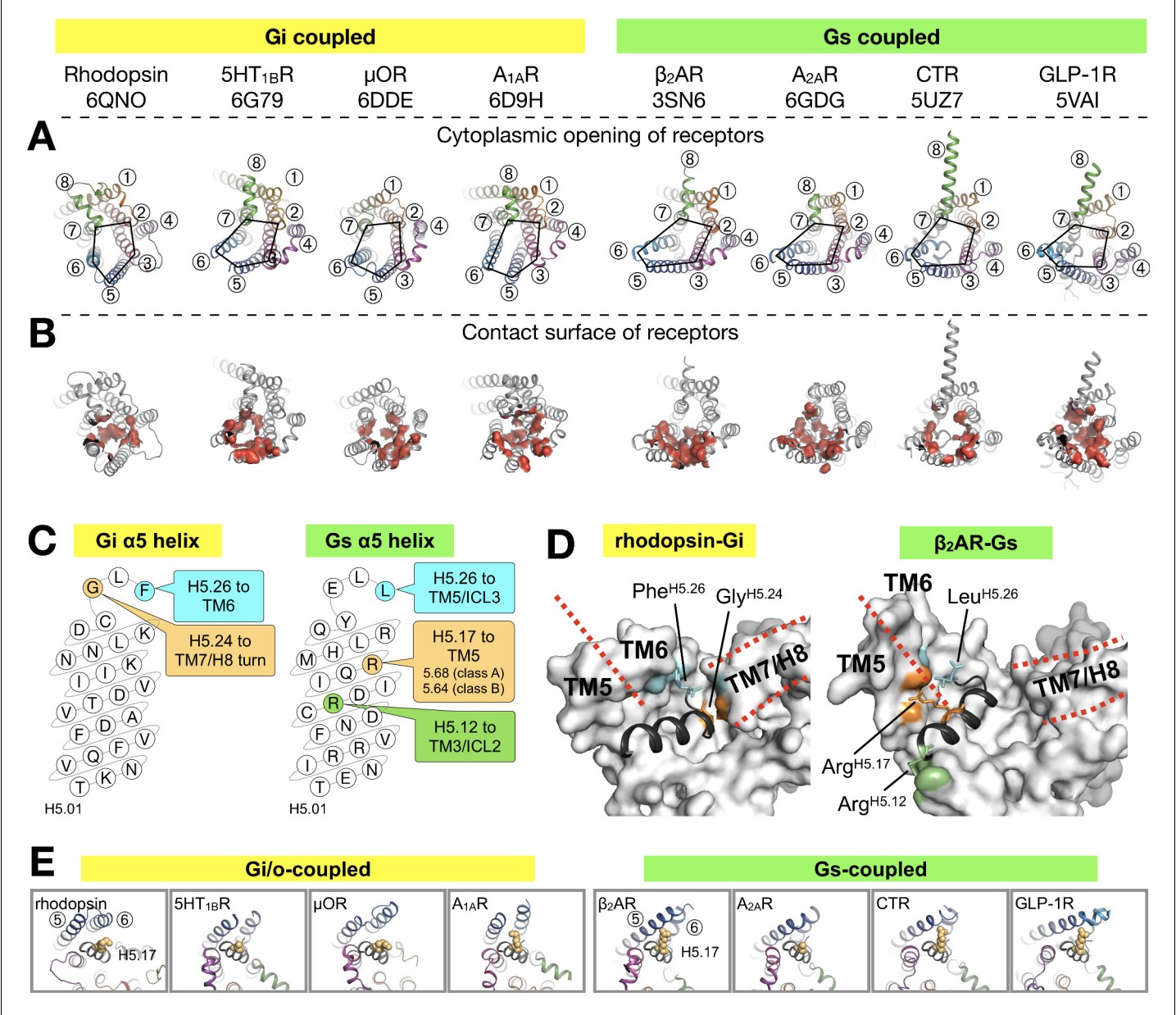

**Figure 3.** Binding of the Gα α5 helix in GPCR-G protein complexes. (**A**) Overview of a subset of GPCR-G protein complexes used for this analysis. For a complete list of the complexes used, see *Figure 3—figure supplement 2*. The complexes are shown from the cytoplasmic side, and the black pentagons connecting TM2-3-5-6-7 delineate the G protein-binding interface. Receptors are represented in ribbons with their TM helices numbered. (**B**) The red surfaces depict the contact area (within a distance of 4 Å) between the receptor and the α5 helix of Gα (**C**) Schematic representation of the α5 helix in Gi- and Gs-subtypes. The residues highlighted, and their respective binding site to the receptor, are conserved among all Gi, and Gs, complexes analyzed. All contacts retrieved from this analysis are reported in *Figure 3—figure supplement 2*. (**D**) Position of Gi- and Gs-specific contacts shown in the rhodopsin-Gi (this work, PDB id: 6QNO) and the β2AR-Gs (PDB id: 3SN6) complexes. The cytoplasmic side of the receptors is depicted as gray surface. Red lines mark the border between TM5 and TM6 and delineate H8. (**E**) Cytoplasmic view of GPCR-G protein complexes showing the interaction between H5.17 (orange spheres) and TM5 and TM6. From left to right, the structures (PDB ids) correspond to: this work, 6G79, 6DDE, 6D9H, 3SN6, 6GDG, 5UZ7, and 5VAI. Throughout the analysis, contacts are defined as atomic distances smaller than 4 Å.

DOI: https://doi.org/10.7554/eLife.46041.016

The following figure supplements are available for figure 3:

**Figure supplement 1.** Residue-residue contact list between GPCRs and Gα H5.

DOI: https://doi.org/10.7554/eLife.46041.017

**Figure supplement 2.** Residue-residue contacts between GPCRs and Gα H5.

DOI: https://doi.org/10.7554/eLife.46041.018

*Figure 3 continued on next page*

*Figure 3 continued*

**Figure supplement 3.** Contacts observed between ICL2/ICL3 and the Gα subunit.
DOI: https://doi.org/10.7554/eLife.46041.019
**Figure supplement 4.** Residue-residue contacts between ICL2/3 and Gα.
DOI: https://doi.org/10.7554/eLife.46041.020
**Figure supplement 5.** Region near ICL2 in available structures.
DOI: https://doi.org/10.7554/eLife.46041.021

identical to our previously reported X-ray structure of the same receptor bound to a mini-Go protein (*Tsai et al., 2018*) (*Figure 1—figure supplement 7B*).

However, our EM map shows a density on the Gβ subunit that extends from H8 of the receptor, constituting the proximal segment of the C-terminus (residues G324 to T335) (*Figure 2*). One explanation of why the C-tail is observed in our density map and not in other structures may rely on the nature of the components used to reconstitute the reported GPCR complexes (*Figure 1—figure supplement 8*). Among those, a meaningful comparison may be done with the µ-opioid receptor complex (*Koehl et al., 2018*), which is bound to a shorter version of our antibody Fab16, but in which the C-terminus of the receptor was partially truncated. In the human rhodopsin complex (*Kang et al., 2018*), which contains the full length C-tail, a recombinant Gβγ was used. Thus, our bovine rhodopsin complex contains the unique combination of a full-length C-tail, Gβγ purified from bovine rod outer segments, and Fab16 that may have contributed to trap the transient interaction of the intrinsically disordered C-tail to Gβ. Remarkably, the recently published structure of the M1 muscarinic acetylcholine receptor in complex with G11 and scFv16 (*Maeda et al., 2019*) reveals part of the receptor C-tail bound to the G protein in essentially the same conformation as in our structure (*Figure 2—figure supplement 4*).

The Gβγ subunit engages with a wide range of effectors (*Dupré et al., 2009*; *Khan et al., 2013*), including direct interactions with GPCRs. For instance, these can associate prior to trafficking to the plasma membrane (*Dupré et al., 2006*) where they can remain pre-coupled (*Galés et al., 2005*). Also, according to a sequential fit model (*Herrmann et al., 2004*; *Herrmann et al., 2006*), activated rhodopsin binds to Gβγ assisting to position the Ras domain of Gα into proximity of its binding site to the receptor. This would facilitate folding of the C-terminal α5 helix of this subunit leading to nucleotide exchange (*Dror et al., 2015*; *Flock et al., 2015*; *Kapoor et al., 2009*). After dissociation of the G protein heterotrimer, the Gβ subunit recruits GRKs to the membrane, resulting in the phosphorylation of the receptor C-tail (*Claing et al., 2002*; *Pao and Benovic, 2002*; *Pitcher et al., 1998*) and binding of arrestin (*Goodman et al., 1996*). Interestingly, HEK cells lacking functional Gα subunits (but retaining the native Gβγ) are still able to recruit arrestin, although they fail to activate ERK and whole-cell responses (*Grundmann et al., 2018*).

Our findings provide a structural explanation for these roles of Gβγ in GPCR-mediated signaling. We suggest that the observed interaction between the central part of the receptor C-terminus and Gβγ is involved in localizing the G protein heterotrimer to the active receptor first. After dissociation of the Gα subunit, a transient Gβγ/receptor complex could provide the adequate molecular context to allow receptor phosphorylation by bringing the kinase close to the receptor C-tail (*Ribas et al., 2007*).

In other class A GPCRs, interactions with Gβγ have been also located in ICL3 (in M2 and M3) muscarinic receptors, involved in the phosphorylation of this loop (*Wu et al., 1998*) and potentially in ICL1 (in A$_{2}$AR and β$_{2}$AR) (*García-Nafría et al., 2018a*). In our complex, K6612.48 and K6712.49 in ICL1 are indeed close to D312 in Gβ (7 Å between Cα atoms); however, we do not observe defined density for the side chains in this region (*Figure 1—figure supplement 6A*). Interestingly, the complexes of the class B GPCRs calcitonin and GLP-1 feature an extended and tilted H8 resulting in extensive contacts with Gβγ that have not been observed in Class A receptors (*García-Nafría et al., 2018a*). H8 has also been shown to bind Gβγ in the class B PTH1 receptor (*Mahon et al., 2006*). Thus, it is likely that class A and class B GPCRs use distinct mechanisms to engage Gβγ.

The receptor-Gβγ interaction observed in our structure is compatible with proposed models of receptor-GRK recognition (*Komolov et al., 2017*; *Sarnago et al., 2003*). Thus, we suggest that the Gβγ subunit can 'tag' activated GPCRs and provide an interface for bringing different effectors (Gα subunits first, and GRKs later) near the cytoplasmic domains of GPCRs.

The availability of several GPCR-G protein complexes has greatly advanced our understanding on how receptors activate the Gα subunit (*Dror et al., 2015*; *Flock et al., 2015*). The binding interface of the receptor is partially formed by ICL2 and ICL3 (*Chung et al., 2011*; *Glukhova et al., 2018*; *Sun et al., 2015*). Accordingly, our EM map shows that these domains are in close proximity to G (*Figure 3—figure supplement 3*). In particular, we found that ICL2 is at contact distance to the αN helix, the αN/β1 and the β2/β3 Gα in most structures (*Figure 3—figure supplements 3–5*). While ICL2 contributes to the binding interface, there are no apparent conserved contacts among complexes (*Figure 3—figure supplement 4*). However, at the interface between receptor and the α5 helix of the G protein, in all Gi complexes Phe$^{H5.26}$ and Gly$^{H5.24}$ contact TM6 and TM7/H8 of the receptor, respectively. Instead, in all Gs complexes we find that Leu$^{H5.26}$, Arg$^{H5.17}$ and Arg$^{H5.12}$ contact TM5/ICL3, TM5, and TM3/ICL2, respectively. In a recent analysis (*Glukhova et al., 2018*), it was proposed that residues 5.68 (class A GPCRs) and 5.64 (class B GPCRs) are particularly important for Gs protein binding. We observe that residue 5.68 interacts with H5.16 in both Gi- and Gs-bound complexes, but only with Arg$^{H5.17}$ in all Gs complexes (*Figure 3—figure supplement 2*). While our analysis is limited by the number of available structures, we suggest that the conserved contacts identified between α5 and the receptors are structurally important anchoring points, but we cannot exclude that they might be relevant at the level of recognition between G protein subtypes.

In order to visualize snapshots of the signaling pathway that have not been observed yet, more work on the structural characterization of complexes is still needed, ideally using the same receptor and different transducers. Such complexes are of pivotal importance to decipher the structural basis of the GPCR-mediated signaling cascade.

# Materials and methods

**Key resources table**

| Reagent type (species) or resource | Designation | Source or reference | Identifier | Addition information |
|---|---|---|---|---|
| Strain, strain background (*E. coli*) | BL21(DE3) | Sigma-Aldrich | SA: CMC0014 | |
| Cell line (*H. sapiens*) | HEK293 GnTI⁻ | Reeves laboratory (gift) | Under MTA with MIT, Cambridge, MA, US. | PMID: 12370423 |
| Cell line (*M. musculus*) | Hybridoma | Hoffmann-La Roche (by collaboration) | Under MTA with Hoffmann-La Roche, Basel, Switzerland. | *Maeda et al., 2018* |
| Biological sample (*B. taurus*) | Bovine retinae | WL Lawson Company (USA) | | https:// wllawsoncompany.com/ |
| Chemical compound, drug | Blue Sepharose 6 Fast Flow | GE Healthcare | GEH: 17094801 | |
| Chemical compound, drug | Chelating Sepharose Fast Flow immobilized metal affinity chromatography resin | GE Healthcare | GEH: 17057501 | |
| Chemical compound, drug | HiTrap Protein G Sepharose HP column | GE Healthcare | GEH: 17040401 | |
| Chemical compound, drug | Immobilized papain agarose resin | Thermo Scientific | TS: 20341 | |

*Continued on next page*

*Continued*

| Reagent type (species) or resource | Designation | Source or reference | Identifier | Addition information |
|---|---|---|---|---|
| Chemical compound, drug | Protein A Sepharose | GE Healthcare | GEH: 17078001 | |
| Chemical compound, drug | CNBr-activated sepharose 4B | GE Healthcare | GEH: 17043001 | |
| Chemical compound, drug | Ultra-low IgG fetal bovine serum | Gibco | Gibco: 16250078 | |
| Chemical compound, drug | Interleukin-6 recombinant mouse protein | Invitrogen | Invitrogen: PMC0064 | |
| Chemical compound, drug | cOmplete EDTA-free protease inhibitor cocktail | Sigma-Aldrich | SA: 1873580001 | |
| Chemical compound, drug | Dodecyl β-maltoside | Anatrace | Anatrace: D310 | |
| Chemical compound, drug | Lauryl-maltoside neopentyl glycol | Anatrace | Anatrace: NG310 | |
| Chemical compound, reagent | 1D4 antibody | Cell Essentials Inc | | http://www.cell-essentials.com/ |
| Chemical compound, drug | 1D4 peptide | Peptide 2.0 | | https://www.peptide2.com/ |
| Chemical compound, drug | 9-cis retinal | Sigma-Aldrich | SA: R5754 | |
| Chemical compound, drug | Apyrase | New England Biolabs | NEB: M0398 | |
| Software, algorithm | XDS | | http://xds.mpimf-heidelberg.mpg.de | PMID: 20124692; RRID: SCR_015652 |
| Software, algorithm | PHENIX | | https://www.phenix-online.org | *Adams et al., 2010*; RRID: SCR_014224 |
| Software, algorithm | UCSF Chimera | | https://www.cgl.ucsf.edu/chimera | *Pettersen et al., 2004*; RRID: SCR_004097 |
| Software, algorithm | PyMOL | Schrödinger LLC | https://pymol.org | RRID:SCR_000305 |
| Software, algorithm | FOCUS | | https://focus.c-cina.unibas.ch/about.php | *Biyani et al., 2017* |
| Software, algorithm | MotionCor2 | | http://msg.ucsf.edu/em/software/motioncor2.html | *Zheng et al., 2017*; RRID: SCR_016499 |
| Software, algorithm | cryoSPARC | Structura Biotechnology Inc. | https://cryosparc.com | PMID: 28165473; RRID: SCR_016501 |
| Software, algorithm | RELION 2, 3 | | http://www2.mrc-lmb.cam.ac.uk/relion | RRID:SCR_016274 |
| Software, algorithm | NAMD | | http://www.ks.uiuc.edu/Research/namd | RRID:SCR_014894 |
| Software, algorithm | Coot | | https://www2.mrc-lmb.cam.ac.uk/personal/pemsley/coot | *Emsley and Cowtan, 2004*; RRID: SCR_014222 |

## Protein expression and purification

The N2C/M257Y/D282C mutant of bovine rhodopsin was expressed as described (*Deupi et al., 2012*) in human embryonic kidney (HEK) 293 GnTI-deficient cells (gift from Philip Reeves), which retain all post-translational modifications of the native protein including palmitoylation of Cys residues at positions 322 and 323 (*Standfuss et al., 2011*). Cell lines were tested and confirmed to be mycoplasma-free. The human Gαi subunit (Gαi1) with an N-terminal TEV protease-cleavable deca-histidine tag was expressed in the *E. coli* BL21 (DE3) strain (Sigma: CMC0014) and purified as described (*Sun et al., 2015*). The transducin heterotrimer was isolated from the rod outer segment of bovine retina (W L Lawson Company) and Gβ1γ1 was separated from Gαt with Blue Sepharose 6 Fast Flow (GE Healthcare) as described (*Maeda et al., 2014*). The Gαi1β1γ1 heterotrimer (Gi) was prepared by mixing equimolar amounts of Gαi1 with or without 10xHis-tag and Gβ1γ1 and incubated at 4°C for 1 hr shortly before use for rhodopsin-Gi complex formation on the 1D4 immunoaffinity column.

## Fab16 production

The monoclonal mouse antibody IgG16 was generated as described (*Koehl et al., 2018*). Large scale production of IgG16 was performed using adherent hybridoma cell culture (Hoffmann-La Roche, collaboration) grown in DMEM medium supplemented with 10% ultra-low IgG fetal bovine serum (FBS) (Gibco, #16250078) and 25 U/ml of mouse interleukin-6 (Invitrogen, #PMC0064) at 37°C and 5% $CO_2$. Antibody expression was increased by stepwise dilution of FBS concentration down to 2%. After incubation for 10–14 days,~500 ml cell suspension containing the secreted IgG was clarified by centrifugation and subsequent filtration through a 0.45 μm HAWP membrane (Merck Millipore). The filtrate was mixed with an equal volume of binding buffer (20 mM $Na_2HPO_4$/$NaH_2PO_4$, pH 7.0) and loaded to a 1 ml HiTrap Protein G Sepharose FF column (GE Healthcare). The column was washed with binding buffer until the $UV_{280}$ absorbance dropped to a stable baseline, and IgG was eluted with 0.1 M glycine-HCl (pH 2.7). Fractions were immediately neutralized with 1 M Tris-HCl (pH 9.6). Fractions containing IgG16 were combined and dialyzed against 20 mM $Na_2HPO_4$/$NaH_2PO_4$ (pH 7.0), 1.5 mM $NaN_3$ using a slide-A-lyzer dialysis cassette (12–14 kDa MWCO, Thermo Scientific) at 4°C for 15 hr. The dialysate was collected and mixed with the immobilized papain resin (0.05 ml resin for 1 mg IgG) (Thermo Scientific, #20341). Papain was activated by the addition of L-cysteine and EDTA to a final concentration of 20 mM and 10 mM respectively. IgG was digested overnight at 37°C with gentle mixing. Afterwards the immobilized papain resin was removed and the digested fraction was mixed with Protein A Sepharose (0.2 ml resin for 1 mg digested IgG, GE Healthcare, #17078001) for 1 hr at RT. Resins were washed with two column volumes (CV) of wash buffer 10 mM Tris (pH 7.5), 2.5 M NaCl. The flow-through and washing fractions containing Fab16 were collected and dialyzed against PBS supplemented with 1.5 mM $NaN_3$ using a slide-A-lyzer dialysis cassette (12–14 kDa MWCO) at 4°C. The dialysate was collected and concentrated with a Viva-Spin 20 concentrator (10 kDa MWCO, Sartorius) to approximately 1.1 mg/ml. Glycerol was supplemented to the concentrated Fab16 at a final concentration of 10%, and protein was flash frozen in liquid nitrogen and stored at −20°C until use for formation of the rhodopsin-Gi-Fab16 complex.

## Fab16 crystallization and structure determination

For its crystallization, Fab16 was further purified by SEC on a Superdex 200 Increase 10/300 GL column (GE Healthcare) equilibrated in buffer 10 mM Tris (pH 7.5), 50 mM NaCl. Fractions containing pure Fab16 were collected and concentrated to approximately 14 mg/ml using a VivaSpin four concentrator (10 kDa MWCO, Sartorius). Fab16 was crystallized by vapor diffusion at 4°C by dispensing 200 nl of protein and 200 nl of crystallization buffer containing 1.5 M malic acid (pH 7.5), 7% (v/v) LDAO in an MRC 2-well crystallization plate (Swissci) using a mosquito crystallization robot. Crystals appeared after one day and grew to full size within 4 days. Crystals were soaked in the reservoir solution supplemented with 20% (v/v) PEG 400 as a cryo-protectant and flash frozen in liquid nitrogen. The X-ray data was collected at the PXI beam line at the Swiss Light Source (SLS). Bragg peaks were integrated using XDS for individual datasets. XSCALE was used to scale and combine six datasets, and the pooled reflection list was further analyzed using the STARANISO server (Global Phasing Ltd.). The STARANISO server first analyzed the anisotropy for each dataset, giving a resolution

of 1.90 Å in overall, 1.90 Å in the *0.92 a\* - 0.38 c\* direction*, 2.25 Å in the *k* direction, and 2.13 Å in the *−0.14 a\* + 0.99 c\* direction* under the criterion of CC½=0.3., following by scaling and merging of the reflections. The light chain from PDB id: 1MJU and the heavy chain from PDB id 2AAB without the CDR regions were used for molecular replacement using the program *Phaser-MR* in the Phenix suite. The model was built automatically using the Phenix *AutoBuild*, and the coordinates were manually adjusted using the visualization program Coot. The structure was refined using the *phenix.refine* to 1.90 Å (*Supplementary file 1*). The structure factor and the coordinates are deposited to the Protein Data Bank under the accession code 6QNK.

## Purification of the rhodopsin-Gi-Fab16 and the rhodopsin-Gi complexes

Buffers for every purification step were cooled to 4°C before use, and the steps after adding retinal were performed under dim red light before light activation of rhodopsin. The stabilized, constitutively active rhodopsin mutant N2C/M257Y/D282C was expressed in HEK293 GnTI⁻ deficient cells as described (*Deupi et al., 2012*). The cells were collected from the cell culture by centrifugation and homogenized in PBS buffer with cOmplete EDTA-free protease inhibitor (Roche). The cells were solubilized by supplementing dodecyl maltoside (DDM) (Anatrace, Sol-grade) to final concentrations of cell at 20% (w/v) and of DDM at 1.25% (w/v). After gentle stirring for 1–2 hr, the solubilized fraction was collected after centrifugation at 200,000x g for 1 hr. Solubilized rhodopsin apoprotein was captured in batch using the immunoaffinity Sepharose beads (GE Healthcare, #1043001) coupled to 1D4 antibody for more than 4 hr at the ratio of 5 g cells per ml resin. 1D4 resins were collected and washed with 10 CV of PBS, 0.04% DDM. Afterwards, resins were resuspended in 2 CV of PBS, 0.04% DDM and 75 µM of 9-cis (Sigma) or 11-cis retinal (Roche, or from NIH) for at least 6 hr in the dark. Later resins were washed with 30 CV of buffer A containing 20 mM HEPES pH 7.5, 150 mM NaCl, 1 mM MgCl$_2$, 0.02% (w/v) lauryl-maltose neopentyl glycol (LMNG) to wash off unbound retinal. Washed resins were resuspended in 1 CV of buffer A and mixed with 10xHis-tagged Gi heterotrimer at the ratio of 2 mg Gi per ml resin with the presence of 25 mU/ml apyrase (New England Biolabs). The resuspended resin/Gi mixture was subjected to irradiation for 10–15 min with light that had been filtered through a 495 nm long-pass filter to induce activation of rhodopsin and G protein binding, followed by a 30 min incubation in the dark to allow full hydrolysis of nucleotide by apyrase. Resins were washed with 8 CV of buffer A to remove unbounded Gi heterotrimer. rhodopsin-Gi complex was eluted three times in batch with 1.5 CV of buffer containing 20 mM HEPES (pH 7.5), 150 mM NaCl, 1 mM MgCl$_2$, 0.02% LMNG, 80 µM 1D4 peptide (TETSQVAPA) for at least 2 hr incubation each time. Elution fractions were combined and incubated for at least 2 hr with Ni-NTA resins (0.5–2 ml), washed with 6 CV of 20 mM HEPES (pH 7.5), 150 mM NaCl, 50 mM imidazole, 0.01% LMNG to remove free rhodopsin. Rhodopsin-Gi was eluted five times with 1 CV of 20 mM HEPES (pH 7.5), 150 mM NaCl, 350 mM imidazole, 0.01% LMNG. Elution fractions were immediately concentrated using an Amicon Ultra concentrator (MWCO 100 kDa) with simultaneous buffer exchange to 20 mM HEPES (pH 7.5), 150 mM NaCl, 0.01% LMNG. Rhodopsin-Gi was mixed with molar excess (1:1.4) of Fab16 and incubated for at least 1 hr. The mixture of rhodopsin-Gi and Fab16 was concentrated using an Amicon Ultra concentrator (MWCO 30 kDa) and loaded to a Superdex 200 Increase 10/300 GL column for SEC with detergent-free buffer containing 20 mM HEPES (pH 7.5), 150 mM NaCl. Protein quality of each fraction was evaluated by UV-VIS measurement and SDS-PAGE. Fractions showing OD$_{280}$/OD$_{380}$ = ~5.9 (ratio between 280 and 380 nm) was chosen for cryo-EM studies. For preparation of rhodopsin-Gi complex, purified rhodopsin and 10xHis-tag-free Gi heterotrimer were mixed in a test tube in equimolar ratio with the presence of 25 mU/ml of apyrase and incubated for 1 hr at 4°C. The protein mixture was concentrated and purified using a Superdex 200 Increase 10/300 GL column. Fractions showing OD$_{280}$/OD$_{380}$ = 3 were used to prepare cryo-EM specimens.

## Cryo-electron microscopy and image analysis

Purified samples of rhodopsin-Gi with and without Fab16 were plunge-frozen in a Vitrobot MarkIV (FEI Company) operated at 4°C and 90% humidity. A drop of 3.5 µL sample at 0.2 mg/mL was dispensed onto a glow discharged lacey carbon grid (Ted Pella, Inc) and blotted for 2–3 s before vitrification in liquid ethane. Images were acquired by a Titan Krios microscope operated at 300 kV equipped with a Falcon III, or with a K2 Summit and GIF energy filter (*Supplementary file 1*).

Datasets were pre-processed and pruned in FOCUS (*Biyani et al., 2017*) using MotionCor2 (*Zheng et al., 2017*) with dose weighting for movie alignment, and CTFFIND4 (*Rohou and Grigorieff, 2015*) for micrograph contrast transfer function estimation. Automated particle picking was performed in Gautomatch (http://www.mrc-lmb.cam.ac.uk/kzhang/Gautomatch/) and all further processing steps were carried in cryoSPARC (Structura Biotechnology Inc) and RELION 2 and 3 (*Kimanius et al., 2016*; *Zivanov et al., 2018*). Around 115,000 particles from the three best resolved 3D classes were pooled and subjected to 3D auto-refinement with a soft mask which deliberately excludes the density of the AH domain observed in one of the 3D classes. Map sharpening and modulation transfer function (MTF) correction were performed with RELION post-processing. The resulting map has a nominal resolution of 4.38 Å estimated following the gold standard Fourier Shell Correlation (FSC) at FSC = 0.143. Local resolution estimation was performed using *blocres* (*Cardone et al., 2013*).

## Model building and structure refinement

The initial models of rhodopsin and the Ras-like domain of Gαi protein were adapted from the structure of rhodopsin-mini-Go complex (PDB id: 6FUF). The initial model of Gβγ was obtained from the crystal structure of guanosine 5'-diphosphate-bound transducin (PDB id: 1GOT). The models were docked into the 3D map as rigid bodies in Chimera (*Pettersen et al., 2004*). After initial refinement using the phenix.real_space_refine in the Phenix suite (*Adams et al., 2010*), the C-tail of rhodopsin was modeled by manually building the residues 324–335 in the unassigned density near H8 and Gβ as a continuation of H8 using Coot (*Emsley and Cowtan, 2004*). The coordinates of the entire structure were improved by further refinement using phenix.real_space_refine and manual adjustment in Coot iteratively. This final placement of the C-tail was then validated by using the molecular dynamics flexible fitting (MDFF) method (*Trabuco et al., 2008*). This technique allows to use electron density data as an external potential added to the molecular mechanics force field, thus taking advantage of the features present in the density while retaining a chemically sound structure. We observed that the modeled C-tail remains stable during the simulation (*Figure 2—figure supplement 3*), supporting a good agreement between the built atomic structure and the density map.

## Structure and sequence analysis

Structural models were downloaded from the Protein Data Bank. For comparing structures of the GPCR-G protein complexes, the residue-residue contacts within 4 Å were first identified using PyMOL (The PyMOL Molecular Graphics System, Version 2.0 Schrödinger, LLC). Electrostatic potentials were calculated using the APBS method (*Baker et al., 2001*) as implemented in PyMOL using a concentration of 0.150 M for the +1 and −1 ion species. The biomolecular surface is colored from red (−5 kT/e) to blue (+5 kT/e) according to the potential on the soluble accessible surface. A sequence alignment of mammalian Gα proteins was obtained from the GPCRdb (*Pándy-Szekeres et al., 2018*). The MDFF simulation (1ns) was performed using Namdinator (*Trabuco et al., 2008*) with default parameters (temperature: 298 K; G-force scaling factor: 0.3; Phenix RSR cycles: 5).

## Figure preparation

Figures were prepared using ChimeraX (*Goddard et al., 2018*) and PyMOL (The PyMOL Molecular Graphics System, Version 2.0 Schrödinger, LLC).

## Data availability

The cryo-EM density map is deposited under accession code EMD-4598 on the EM Data Bank. The related structure coordinates of the rhodopsin-complex bound to Fab16 (accession code 6QNO) and the crystal structure of Fab16 (accession code 6QNK) are deposited on the Protein Data Bank.

## Acknowledgements

We thank ScopeM at ETH Zurich for support, and in particular to Mr. Peter Titmann for help with the operation of Titan Krios for the Falcon three datasets. We are indebted to Dr. Mohamed Chami from the Center for Cellular Imaging and NanoAnalytics in Basel for his help with the initial screening

of cryo-grids. We thank Dr. Takashi Ishikawa and the Electron Microscopy Facility of PSI for his support.We thank Jean-Philippe Carralot (F Hoffmann-La Roche Ltd) for help in antibody generation, Martin Siegrist, Georg Schmid, Bernard Rutten, Doris Zulauf, Stephanie Kueng (Roche Non-Clinical Biorepository) and Ralf Thoma for technical assistance for biomass and cell line generation.

## Additional information

### Competing interests

Hugues Matile, Roger JP Dawson: Employee of Hoffmann-La Roche Ltd. Gebhard Schertler: declares that he is a co-founder and scientific advisor of the company leadXpro AG and InterAx Biotech AG, and that he has been a member of the MAX IV Scientific Advisory Committee during the time when the research has been performed. The other authors declare that no competing interests exist.

### Funding

| Funder | Grant reference number | Author |
| --- | --- | --- |
| Schweizerischer Nationalfonds zur Förderung der Wissenschaftlichen Forschung | 310030_153145 | Gebhard Schertler |
| Swiss Nanoscience Institute | A13.12 NanoGhip | Gebhard Schertler |
| Schweizerischer Nationalfonds zur Förderung der Wissenschaftlichen Forschung | TransCure | Henning Stahlberg |
| Holcim Stiftung | | Jacopo Marino |
| ETH Zürich | | Tilman Flock |
| University of Cambridge | | Tilman Flock |
| Roche | | Shoji Maeda |
| Schweizerischer Nationalfonds zur Förderung der Wissenschaftlichen Forschung | 160805 | Xavier Deupi |
| Schweizerischer Nationalfonds zur Förderung der Wissenschaftlichen Forschung | 310030B_173335 | Gebhard Schertler |

The funders had no role in study design, data collection and interpretation, or the decision to submit the work for publication.

### Author contributions

Ching-Ju Tsai, Formal analysis, Investigation, Visualization, Writing—original draft, Writing—review and editing; Jacopo Marino, Formal analysis, Funding acquisition, Investigation, Visualization, Writing—original draft, Writing—review and editing; Ricardo Adaixo, Filip Pamula, Tilman Flock, Formal analysis, Investigation, Writing—review and editing; Jonas Muehle, Shoji Maeda, Roger JP Dawson, Resources, Investigation, Writing—review and editing; Nicholas MI Taylor, Investigation; Inayatulla Mohammed, Resources; Hugues Matile, Resources, Investigation; Xavier Deupi, Formal analysis, Supervision, Funding acquisition, Writing—original draft, Writing—review and editing; Henning Stahlberg, Supervision, Funding acquisition, Writing—review and editing; Gebhard Schertler, Conceptualization, Supervision, Funding acquisition, Project administration, Writing—review and editing

### Author ORCIDs

Ching-Ju Tsai (iD) https://orcid.org/0000-0001-8320-5009
Jacopo Marino (iD) https://orcid.org/0000-0001-7095-0800
Nicholas MI Taylor (iD) https://orcid.org/0000-0003-0761-4921
Xavier Deupi (iD) https://orcid.org/0000-0003-4572-9316

Henning Stahlberg (iD) https://orcid.org/0000-0002-1185-4592
Gebhard Schertler (iD) https://orcid.org/0000-0002-5846-6810

### Decision letter and Author response

Decision letter https://doi.org/10.7554/eLife.46041.039
Author response https://doi.org/10.7554/eLife.46041.040

## Additional files

### Supplementary files

• Supplementary file 1. Supplementary Tables. Supplementary Table 1. Cryo-EM data collection and refinement statistics. Supplementary Table 2. Crystallographic data and structural refinement of Fab16.

DOI: https://doi.org/10.7554/eLife.46041.022

• Transparent reporting form

DOI: https://doi.org/10.7554/eLife.46041.023

### Data availability

The cryo-EM density map of the rhodopsin-Gi complex bound to Fab16 has been deposited in the EM Data Bank (accession code EMD-4598), and the related structure coordinates have been deposited in the Protein Data Bank (accession code 6QNO). The crystal structure of Fab16 has been deposited in the Protein Data Bank (accession code 6QNK). Source data for Figure 3 is provided in Suppl. Table 3.

The following datasets were generated:

| Author(s) | Year | Dataset title | Dataset URL | Database and Identifier |
|---|---|---|---|---|
| Tsai CJ, Marino J, Adaixo RJ, Pamula F, Muehle J, Maeda S, Flock T, Taylor NMI, Mohammed I, Matile H, Dawson RJP, Deupi X, Stahlberg H, Schertler GFX | 2019 | Rhodopsin-Gi protein complex | http://www.ebi.ac.uk/pdbe/entry/emdb/EMD-4598 | Electron Microscopy Data Bank, EMD-4598 |
| Tsai CJ, Marino J, Adaixo RJ, Pamula F, Muehle J, Maeda S, Flock T, Taylor NMI, Mohammed I, Matile H, Dawson RJP, Deupi X, Stahlberg H, Schertler GFX | 2019 | Rhodopsin-Gi protein complex | http://www.rcsb.org/structure/6QNO | Protein Data Bank, 6QNO |
| Tsai CJ, Muehle J, Pamula F, Dawson RJP, Maeda S, Deupi X, Schertler GFX | 2019 | Antibody FAB fragment targeting Gi protein heterotrimer | http://www.rcsb.org/structure/6QNK | Protein Data Bank, 6QNK |

The following previously published datasets were used:

| Author(s) | Year | Dataset title | Dataset URL | Database and Identifier |
|---|---|---|---|---|
| Tsai CJ, Weinert T, Muehle J, Pamula F, Nehme R, Flock T, Nogly P, Edwards PC, Carpenter B, Gruhl T, Ma P, Deupi X, Standfuss J, Tate CG, | 2018 | Crystal structure of the rhodopsin-mini-Go complex | https://www.rcsb.org/structure/6FUF | Protein Data Bank, 6FUF |

Schertler GFX

| | | | | |
|---|---|---|---|---|
| Lambright DG, Sondek J, Bohm A, Skiba NP, Hamm HE, Sigler PB | 1997 | Heterotrimeric complex of a Gt-alpha/Gi-alpha chimera and the Gt-beta-gamma subunits | https://www.rcsb.org/structure/1GOT | Protein Data Bank, 1GOT |
| Chang CC, Hernandez-Guzman FG, Luo W, Wang X, Ferrone S, Ghosh D | 2005 | Structural basis of antigen mimicry in a clinically relevant melanoma antigen system | https://www.rcsb.org/structure/2AAB | Protein Data Bank, 2AAB |
| Zhu Y, Wilson IA | 2002 | Crystal structure of murine class II MHC I-Ab in complex with a human CLIP peptide | https://www.rcsb.org/structure/1MUJ | Protein Data Bank, 1MUJ |

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
