## [Decision Letter]

Thank you for submitting your article "Cryo-EM structure of the rhodopsin-Gαi-βγ complex reveals binding of the rhodopsin C-terminal tail to the Gβ subunit" for consideration by *eLife*. Your article has been reviewed by three peer reviewers, including Nikolaus Grigorieff as the Reviewing Editor and Reviewer #1, and the evaluation has been overseen by Richard Aldrich as the Senior Editor. The following individual involved in review of your submission has agreed to reveal their identity: Dan Oprian (Reviewer #3).

The reviewers have discussed the reviews with one another and the Reviewing Editor has drafted this decision to help you prepare a revised submission.

Summary:

The authors describe the cryo-EM structure of a rhodopsin-Gαi-βγ complex, bound to a Fab fragment to aid structure determination. The structure reveals several important aspects of GPCR-G protein interactions. The described interactions between the C-terminal helix of Gαi (H5) with a network of residues on TMs 3, 5, 6 and 7/H8 is consistent with previous GPCR-G protein complexes and true for all receptor and G protein isoform. The interaction of ICL2 with αN/β1 of Gαi is also consistent with previous receptor-G protein complexes. The present structure offers insight into the diversity of the GPCR-G protein interactions since it provides a direct comparison of rhodopsin-Gi and rhodopsin-Gt complexes. The authors give a nice summary and analysis of the GPCR-G protein interfaces, comparing and contrasting recently reported cryo-EM structures of the GPCR-G protein complexes. What distinguishes this structure from previous complexes is an apparent interaction of the rhodopsin C-terminus with Gβ. This "C-tail", previously reported as unstructured in either the inactive rhodopsin structure or in complex with G proteins, appears to migrate away from the putative membrane toward Gαi-Gβ interface.

Overall, the structure provides insight into the diversity of the GPCR-G protein interactions and highlights a novel interaction between the rhodopsin C-tail and the heterotrimer. It will serve as a guide to future work, which will no doubt be undertaken by multiple groups.

Essential revisions:

The following main concerns should be addressed by the authors:

1) The rhodopsin C-tail interaction with the G protein is one of the main discoveries of this work. The manuscript would be strengthened if the authors could discuss in more depth the role of the residues involved in this interaction, particularly on the G protein (both subunits). Are they important for G protein recruitment and activation? Is there evidence, other than the new structure presented here, to validate the model?

2) Details of the final density map shown in Figure 1—figure supplements 6, and 7 and Figure 3—figure supplement 3 suggest that map features are weak or ambiguous in some places. Especially density for the C-tail and intracellular loop 2 appears to be weak. Since the main conclusions of the work are based on these regions, the authors should describe in more detail how they built the model in these regions and how they characterized potential errors.

3) Helix 8 of rhodopsin in the Gi complex is only composed of two turns whereas in all other rhodopsin structures (inactive Rho, metaRho, opsin, Rho-Gt, and Rho-arrestin) it appears to have three turns. A 3-turn helix may be expected when it is capped by palmitoylation at C322/C323. Is the 2-turn helix observed by the authors the result of depalmitoylation? Allowing the C-terminus to unravel by one turn may allow it to extent into the Gα-Gβ interface. Perhaps one of the reasons why a C-tail interaction with the G protein has not been observed in other rhodopsin complexes is the palmitoylation of the cysteines. Previous studies have suggested that depalmitoylated rhodopsin display normal G protein coupling but the receptor is quite unstable and can result in various retinopathies. Is it possible that the partial unraveling of the helix 8, and the resulting novel interactions, are the result of depalmitoylation? What can the authors say about the palmitoylation state of rhodopsin in their complex?

4) Is it possible that the structure of helix 8 is affected by the detergent used to reconstitute the complex? In a membrane, the 3-turn helix 8 may be stabilized owing to its amphipathicity. This should be discussed by the authors.

Reviewer #1:

The authors describe the cryo-EM structure of a rhodopsin-Gαi-βγ complex, bound to a Fab fragment to aid in image processing. They built a model into the map, including the C-terminal tail of rhodopsin, which appears to interact with the Gβ subunit. The C-tail has not been observed in previous rhodopsin-G protein structures and the authors suggest that the interactions they observe are important for specific binding of G proteins to the receptor. Comparing their structure with other GPCR structures, the authors show that certain amino acid residues make conserved contacts between the receptor and G proteins that are subtype specific.

Mapping the specific contacts between the receptor and the G protein heterotrimer is important for understanding the molecular mechanism of GPCR signaling pathways, and the present work represents, therefore, an important advance in the field. However, details of the final density map shown in Figure 1—figure supplements 6 and 7 and Figure 3—figure supplement 3 suggest that map features are weak or ambiguous in some places. Especially density for the C-tail and intracellular loop 2 appears to be weak. Since the main conclusions of the work are based on these regions, the authors should describe in more detail how they built the model in these regions and how they characterized potential errors.

Reviewer #2:

The CryoEM structure of a complex between rhodopsin and heterotrimeric Gai•Gbg reveals several important aspects of GPCR-G protein interactions. Among these, interactions between the C-terminal helix of Gαi (α5, or H5) with a network of residues on TMs 3, 5, 6 and 7/H8 is consistent with previous GPCR-G protein complexes, regardless of the receptor or G protein isoform. The interaction of ICL2 with the αN-β1 of Gαi is also consistent with previous receptor-G protein complexes. The structure does provide insight into the diversity of the GPCR-G protein interactions since it provides a direct comparison of Rho-Gi and Rho-Gt complexes. The authors provide a nice summary and analysis of the GPCR-G protein interfaces comparing and contrasting recently reported cryoEM structures of the GPCR-G protein complexes. What distinguishes this structure from previous complexes is an apparent interaction of the rhodopsin C-term with Gβ. In this current structure the C-terminal domain, previously reported as unstructured in either the inactive rhodopsin structure or in complex with G proteins, appears to migrate away from the putative membrane toward Gαi-Gβ interface.

Overall, the structure does provide some additional insight into the diversity of the GPCR-G protein interactions and highlights a possible novel interaction with the Rho C-term and the heterotrimer. However, there is really a lack of experimental data to both validate the structural model and also to provide some functional relevance of this potentially interesting interaction.

Rho C-term-G protein interaction: The Rho C-terminal interaction with the G protein would represent a novel interaction. It is surprising that the authors did not devote more attention in the manuscript to highlighting and characterizing the precise role of the residues involved in the interaction, particularly on the G protein (both subunits). Moreover, like any other structure, the model needs to be tested in some functional assay. Some attempt to validate the structure and the C-terminal interaction with the G protein heterotrimer should be made, at least to determine its physiological or pathological relevance. Site-directed mutagenesis of candidate interacting residues and examining their role in G protein recruitment and activation would be logical studies.

From the models provided it appears that Helix 8 of rhodopsin in the Gi complex is only composed of two turns whereas Helix 8 in all other rho structures (inactive Rho, metaRho, opsin, Rho-Gt, and Rho-arrestin) appear to have a three-turn helix. One of the reasons a 3-turn helix is logical is that helix 8 of Rho is capped by palmitoylation at C322/C323. This would suggest that perhaps the Rho isolated in the complex described in this current study is depalmitoylated. Allowing the C-term to unravel by one turn may allow the C-term to extent into the Gα-Gβ interface. Perhaps one of the reasons why such interactions with other Rho-G protein structures have not been observed is that their palmitoylated cysteines may be intact. Previous studies have suggested that depalmitoylated Rho appears to display normal G protein coupling but the receptor is quite unstable and can result in various retinopathies. In this current study it would suggest that unraveling of the helix 8, as a result of depalmitoylation may allow for unique rho-G protein interactions. If this is the case then the palmitoylation state of rho should be tested.

A complicating factor is that helix 8 may be influenced by the detergent environment used in the sample preparation and EM imaging. In a membrane environment the entire 3-turn helix 8 may be stabilized owing to its amphipathicity. In detergent micelles, and particularly if helix 8 is depalmitoylated, it is likely that the influence of amphipathicity may be negated and thus allow helix unraveling.

Reviewer #3:

The manuscript by Tsai et al. entitled "Cryo-EM structure of the rhodopsin-Gαi-βγ complex reveals binding of the rhodopsin C-terminal tail to the Gβ subunit" is a well-written description of a study in which the authors determine the three-dimensional structure of a rhodopsin-G protein complex to a nominal global resolution of 4.38 Å using cryo-electron microscopy. The major new insight from this study is the observation of density extending from H8 of the receptor into the Gβ subunit of the heterotrimeric G protein, a feature not observed in previous structures and one which the authors use to rationalize interactions of the receptor with not only the G protein but also other signaling partners such as the receptor kinases. The experiments are well designed, the results are clear-cut, and the conclusions rest soundly on the results. The study is timely and will appeal to a broad readership.

---

## [Author Response]

Essential revisions:The following main concerns should be addressed by the authors:1) The rhodopsin C-tail interaction with the G protein is one of the main discoveries of this work. The manuscript would be strengthened if the authors could discuss in more depth the role of the residues involved in this interaction, particularly on the G protein (both subunits). Are they important for G protein recruitment and activation? Is there evidence, other than the new structure presented here, to validate the model?

We thank the reviewers for pointing out this oversight from our side, as we indeed did not discuss the residues in the Gα and Gβγ subunits involved in the interaction with the receptor C-tail. The revised text now includes a more detailed discussion of these residues, including a new panel D in Figure 2 depicting these interactions, and new supplementary figures showing the electrostatic potentials on the receptor/Gα/Gβ interface (Figure 2—figure supplement 1) and the sequence conservation of the interacting residues in Gα and Gβγ (Figure 2—figure supplement 2).

“In our G protein complex, the C-tail stretches over a polar surface on the cleft between Gα and Gβ, interacting with both subunits (Figure 2C and D). […] Interestingly, these regions in Gα and Gβ that contact the receptor C-tail are also involved in recognition of GRK2 (Tesmer, et al., 2005).”

A recently published cryo-EM structure of the M1 muscarinic acetylcholine receptor in complex with G11 (Maeda et al., 2019) also reveals the proximal part of the receptor C-tail (11 residues), which binds the G protein in essentially the same conformation as in our structure. This information has also been included in the revised text.

“The EM map reveals a density on the Gβ subunit as continuation of H8 of the receptor (Figure 2A), which corresponds to the C-tail of rhodopsin. This feature has only been observed in the recent structure of the M1 muscarinic acetylcholine receptor in complex with G11 (Maeda et al., 2019).”

2) Details of the final density map shown in Figure 1—figure supplements 6, and 7 and Figure 3—figure supplement 3 suggest that map features are weak or ambiguous in some places. Especially density for the C-tail and intracellular loop 2 appears to be weak. Since the main conclusions of the work are based on these regions, the authors should describe in more detail how they built the model in these regions and how they characterized potential errors.

In the ‘Materials and methods’ section of the revised manuscript, we describe in more detail the modeling process of the rhodopsin C-tail.

“After initial refinement using the phenix.real_space_refine in the Phenix suite (Adams et al., 2010), the C-tail of rhodopsin was modeled by manually building the residues 324-335 in the unassigned density near H8 and Gβ as a continuation of H8 using Coot (Emsley and Cowtan, 2004). […] We observed that the modeled C-tail remains stable during the simulation (Figure 2—figure supplement 3), supporting a good agreement between the built atomic structure and the density map.”

3) Helix 8 of rhodopsin in the Gi complex is only composed of two turns whereas in all other rhodopsin structures (inactive Rho, metaRho, opsin, Rho-Gt, and Rho-arrestin) it appears to have three turns. A 3-turn helix may be expected when it is capped by palmitoylation at C322/C323. Is the 2-turn helix observed by the authors the result of depalmitoylation? Allowing the C-terminus to unravel by one turn may allow it to extent into the Gα-Gβ interface. Perhaps one of the reasons why a C-tail interaction with the G protein has not been observed in other rhodopsin complexes is the palmitoylation of the cysteines. Previous studies have suggested that depalmitoylated rhodopsin display normal G protein coupling but the receptor is quite unstable and can result in various retinopathies. Is it possible that the partial unraveling of the helix 8, and the resulting novel interactions, are the result of depalmitoylation? What can the authors say about the palmitoylation state of rhodopsin in their complex?

Regarding the structure of helix 8, while we observe density up to residue L321(8.58) (Figure 1—figure supplement 6A), the density weakens around residues C322 and C323. However, we do not expect that rhodopsin is depalmitoylated, as our construct was expressed and handled in conditions that have been shown to retain all post-translational modifications (Standfuss et al., 2011; Deupi et al., 2012; Singhal. et al., EMBO Rep. 2013, 14, 520–526; Singhal et al., EMBO Rep. 2016, e201642671–10). While there is indeed local flexibility or structural heterogeneity at the palmitoylation site, possibly due to an impaired interaction of the palmitoyl with the detergent micelle (Glukhova et al., 2018), we expect that helix 8 retains an overall structure that resembles the 3-turn helix observed in all rhodopsin structures. Such 3-turn helix 8 allowed us to build the C-tail into the neighboring density, and MDFF simulations (see above) show that this arrangement is stable and that the interaction with the G protein does not require partial unraveling of helix 8, at least in the context of the present structure.

Part of this information is now included in the ‘Materials and methods’ section of the revised manuscript.

“The N2C/M257Y/D282C mutant of bovine rhodopsin was expressed as described (Deupi et al., 2012) in human embryonic kidney (HEK) 293 GnTI− cells, which retain all post-translational modifications of the native protein including palmitoylation of Cys residues at positions 322 and 323 (Standfuss et al., 2011). Cell lines were tested and confirmed to be mycoplasma-free.”

4) Is it possible that the structure of helix 8 is affected by the detergent used to reconstitute the complex? In a membrane, the 3-turn helix 8 may be stabilized owing to its amphipathicity. This should be discussed by the authors.

In this work we used the detergent lauryl maltose-neopentyl glycol (LMNG), which is a common choice in structural determination of GPCR-G protein complexes by cryo-EM (Munk et al., 2019). Among these, helix 8 in the μ-opioid receptor (Koehl et al., 2018) is not well resolved, while its density is clear in the adenosine A1 receptor (Draper-Joyce et al., 2018), in the M1 and M2 muscarinic acetylcholine receptors (Maeda et al., 2019), and, most notably, in the class B calcitonin (Liang et al., Nature 2017, 546, 118–123) and GLP-1 (Liang et al., Nature 2018, 555, 121–125) receptors. Thus, the use of LMNG as a detergent to reconstitute the complex does not seem to be a major factor in the overall structure of helix 8. However, there exists the possibility that the detergent micelle affects locally the structure of helix 8. Indeed, we observe poor density around residues C322 and C323, at the transition to the C-terminus. This is also the more flexible region in our MDFF simulation (Figure 2—figure supplement 3). This local flexibility or structural heterogeneity may arise from impaired interactions of the receptor with the detergent micelle (Glukhova et al., 2018).

We have added this information in the legend of Figure 2A of the revised manuscript, which depicts the poor density around residues C322 and C323.

“Figure 2. The C-terminal tail of rhodopsin. (A) The EM map is contoured at two different levels to show the continuity of the density. […] TM7, H8 and the C-tail of the receptor are colored in blue, Gα in green, Gβ in yellow, and Gγ in magenta.”

Reviewer #1:[…] Mapping the specific contacts between the receptor and the G protein heterotrimer is important for understanding the molecular mechanism of GPCR signaling pathways, and the present work represents, therefore, an important advance in the field. However, details of the final density map shown in Figure 1—figure supplements 6 and 7 and Figure 3—figure supplement 3 suggest that map features are weak or ambiguous in some places. Especially density for the C-tail and intracellular loop 2 appears to be weak. Since the main conclusions of the work are based on these regions, the authors should describe in more detail how they built the model in these regions and how they characterized potential errors.

We thank the reviewer for his praise. His concerns are addressed in point 2 of the “Essential revisions” section above.

Reviewer #2:[…] Overall, the structure does provide some additional insight into the diversity of the GPCR-G protein interactions and highlights a possible novel interaction with the Rho C-term and the heterotrimer. However, there is really a lack of experimental data to both validate the structural model and also to provide some functional relevance of this potentially interesting interaction.Rho C-term-G protein interaction: The Rho C-terminal interaction with the G protein would represent a novel interaction. It is surprising that the authors did not devote more attention in the manuscript to highlighting and characterizing the precise role of the residues involved in the interaction, particularly on the G protein (both subunits).

We thank the reviewer for pointing out this glaring omission in our study. We address this concern in point 1 of the “Essential revisions” section above.

Moreover, like any other structure, the model needs to be tested in some functional assay. Some attempt to validate the structure and the C-terminal interaction with the G protein heterotrimer should be made, at least to determine its physiological or pathological relevance. Site-directed mutagenesis of candidate interacting residues and examining their role in G protein recruitment and activation would be logical studies.

We thank the reviewer for this positive criticism. We believe that this is a structural study, and that the determination of the pathological relevance of the proposed interaction between the C-terminus of the receptor and the G protein is beyond the scope of our work. However, we agree that further validation and characterization of this interaction is required. We are currently using NMR to assess the binding of the rhodopsin C-terminus to the entire heterotrimeric Gi protein complex. This, and additional biophysical and biochemical characterization, will be part of further studies.

Interestingly, the recent publication of the cryo-EM structure of the M1 muscarinic acetylcholine receptor in complex with G11 (Maeda et al., 2019) supports our structure of the C-tail (see point 1 of the “Essential revisions” section above).

From the models provided it appears that Helix 8 of rhodopsin in the Gi complex is only composed of two turns whereas Helix 8 in all other rho structures (inactive Rho, metaRho, opsin, Rho-Gt, and Rho-arrestin) appear to have a three-turn helix. One of the reasons a 3-turn helix is logical is that helix 8 of Rho is capped by palmitoylation at C322/C323. This would suggest that perhaps the Rho isolated in the complex described in this current study is depalmitoylated. Allowing the C-term to unravel by one turn may allow the C-term to extent into the Gα-Gβ interface. Perhaps one of the reasons why such interactions with other Rho-G protein structures have not been observed is that their palmitoylated cysteines may be intact. Previous studies have suggested that depalmitoylated Rho appears to display normal G protein coupling but the receptor is quite unstable and can result in various retinopathies. In this current study it would suggest that unraveling of the helix 8, as a result of depalmitoylation may allow for unique rho-G protein interactions. If this is the case then the palmitoylation state of rho should be tested.

This concerns is addressed in point 3 of the “Essential revisions” section above.

A complicating factor is that helix 8 may be influenced by the detergent environment used in the sample preparation and EM imaging. In a membrane environment the entire 3-turn helix 8 may be stabilized owing to its amphipathicity. In detergent micelles, and particularly if helix 8 is depalmitoylated, it is likely that the influence of amphipathicity may be negated and thus allow helix unraveling.

This concerns are addressed in point 4 of the “Essential revisions” section above.

Reviewer #3:The manuscript by Tsai et al. entitled "Cryo-EM structure of the rhodopsin-Gαi-βγ complex reveals binding of the rhodopsin C-terminal tail to the Gβ subunit" is a well-written description of a study in which the authors determine the three-dimensional structure of a rhodopsin-G protein complex to a nominal global resolution of 4.38 Å using cryo-electron microscopy. The major new insight from this study is the observation of density extending from H8 of the receptor into the Gβ subunit of the heterotrimeric G protein, a feature not observed in previous structures and one which the authors use to rationalize interactions of the receptor with not only the G protein but also other signaling partners such as the receptor kinases. The experiments are well designed, the results are clear-cut, and the conclusions rest soundly on the results. The study is timely and will appeal to a broad readership.

We wholeheartedly thank the reviewer for his compliments. His comments have been incorporated into the revised manuscript.